# Antioxidant and Antimicrobial Activity of Algal and Cyanobacterial Extracts: An In Vitro Study

**DOI:** 10.3390/antiox11050992

**Published:** 2022-05-19

**Authors:** Sara Frazzini, Elena Scaglia, Matteo Dell’Anno, Serena Reggi, Sara Panseri, Carlotta Giromini, Davide Lanzoni, Carlo Angelo Sgoifo Rossi, Luciana Rossi

**Affiliations:** Department of Veterinary Medicine and Animal Sciences—DIVAS, Università degli Studi di Milano, 26900 Lodi, Italy; sara.frazzini@unimi.it (S.F.); elena.scaglia@unimi.it (E.S.); serena.reggi@unimi.it (S.R.); sara.panseri@unimi.it (S.P.); carlotta.giromini@unimi.it (C.G.); davide.lanzoni@unimi.it (D.L.); carlo.sgoifo@unimi.it (C.A.S.R.); luciana.rossi@unimi.it (L.R.)

**Keywords:** algae, antioxidant, growth inhibition, antimicrobial, metabolomics, polyphenols, IPEC-J2, functional feed, *Ascophyllum nodosum*, O138 *E. coli*

## Abstract

Algae and cyanobacteria, other than their nutritional value, possess different beneficial properties, including antioxidant and antimicrobial ones. Therefore, they can be considered functional ingredients in animal feed and natural substitutes for antibiotics. The aim of this study was to evaluate the antioxidant and antimicrobial capacity against porcine O138 *E. coli* of *Ascophyllum nodosum, Chlorella vulgaris, Lithotamnium calcareum, Schizochytrium* spp. as algal species and *Arthrospira platensis* as cyanobacteria. The antioxidant capacity was determined by ABTS Radical Cation Decolorization Assay testing at three different concentrations (100%; 75%; 50%). The growth inhibition effect of the extracts at concentrations of 25%, 12.5%, 6%, 3% and 1.5% against porcine O138 *E. coli* was genetically characterized by PCR to detect the presence of major virulence factors; this was evaluated by following the microdilution bacterial growth method. The ABTS assay disclosed that *Ascophyllum nodosum* was the compound with the major antioxidant properties (57.75 ± 1.44 percentage of inhibition; *p* < 0.0001). All the extracts tested showed growth inhibition activity at a concentration of 25%. Among all extracts, *A. nodosum* was the most effective, showing a significant growth inhibition of *E. coli*; in particular, the log_10_ cells/mL of *E. coli* used as a control resulted in a significantly higher concentration of 25% and 12.5% after 4 h (8.45 ± 0.036 and 7.22 ± 0.025 log_10_ cells/mL, respectively; *p* < 0.005). This also suggests a dose-dependent relationship between the inhibitory activity and the concentration. Also, a synergistic effect was observed on antioxidant activity for the combination of *Ascophyllum nodosum* and *Lithotamnium calcareum* (*p* < 0.0001). Moreover, to determine if this combination could affect the viability of the IPEC-J2 cells under the normal or stress condition, the viability and membrane integrity were tested, disclosing that the combination mitigated the oxidative stress experimentally induced by increasing the cell viability. In conclusion, the results obtained highlight that the bioactive compounds of algal species are able to exert antioxidant capacity and modulate O138 *E. coli* growth. Also, the combination of *Ascophyllum nodosum* and *Lithotamnium calcareum* species can enhance their bioactivity, making them a promising functional feed additive and a suitable alternative to antibiotics.

## 1. Introduction

Antimicrobial resistance (AMR) is one of the most important threats worldwide [1,2]. More and more antibiotic-resistant organisms and new resistance mechanisms are emerging and spreading globally, threatening our ability to treat common infectious diseases. An increasing number of infections are becoming more difficult to treat due to the reduced effectiveness of antibiotics [3]. However, the use of antibiotics remains crucial for the treatment of certain infectious diseases of bacterial origin. Although animals are not the only contributors to the problem of antibiotic resistance, it is necessary to find alternatives that can reduce the use of antibiotic drugs in animal farming.

In this context, a special role is played by pig farming, where antibiotic drugs are often used to cope with critical phases of the pig’s life, such as weaning, where piglets often develop multifactorial diseases that require antibiotic treatments. The most widely prescribed antibiotics in pig farming are those used for the treatment of diseases caused by different pathotypes of *Escherichia coli* during the post-weaning phase [4]. Among the different *E. coli* strains, those belonging to serogroups O138, O139 and O141 are characterized by a virulence profile responsible for acute and severe enterotoxaemia [5,6]. Therefore, it is necessary to search for new sustainable feed additives that can replace antibiotics, ensuring the sustainable development of livestock systems in line with the principles of One Health [7,8,9].

Among all possible feed additives proposed, algae and cynobacteria, given their composition, could be valuable as functional additives. In addition to their nutritional qualities, they are a rich source of many biologically active compounds and one of the richest sources of natural antioxidants and antimicrobial compounds [10].

Algae are the most common organisms in aquatic environments and belong to a complex heterogeneous group in terms of ecological, taxonomic, morphological and biochemical aspects [11,12]. Algae can be divided into two main categories: microalgae and seaweeds. Macroalgae or seaweeds are a heterogenous group of pluricellular marine organisms, capable of adapting to the severe conditions of marine environments by producing unique natural compounds. In particular, they are known for their content of bioactive substances such as polysaccharides, proteins, lipids and polyphenols that contribute to the antibacterial, antiviral and antifungal properties of these organisms [13,14]. Microalgae are a large group of photosynthetic unicellular eukaryotes [15]. They produce a great variety of compounds, among which include polysaccharides, lipids, proteins, carotenoids, pigments, vitamins, sterols, enzymes, antibiotics, pharmaceuticals and some fine chemicals, as well as biofuels [16].

Animal nutrition plays a pivotal role in maintaining animal health. Both microalgae and seaweeds are actually used in animal nutrition. *Ascophyllum nodosum* (L.) Le Jolis is a large, common cold-water seaweed brown alga that belongs to the family of the Fucaceae, due to its content of vitamins, trace elements, lipids, carbohydrates, proteins and iodine. *Ascophyllum nodosum* is one of the most studied seaweeds in animal nutrition [17]. In addition to brown algae, red ones are also known for their nutritional properties, such as protein content. Among these *Lithothamnium calcareum* (Pallas) Areschoug has been successfully used in bovine feed where, in addition to being an alternative protein source, it can modulate the rumen pH [18,19]. Microalgae, such as *Chlorella vulgaris* var. *vulgaris* Beijerinck, could be a promising ingredient for animal nutrition along with macroalgae; to date, microalgae have found a number of industrial applications. They are used in animal feed due to their content of high-quality proteins, vitamins, carotenoids and n-3 [20,21]. Recent studies have shown that cyanobacteria, such as *Arthrospira platensis* (Nordstedt) Gomont, also positively influence the physiology of animals, improving their immune response and fertility and maintaining better weight control when incorporated in the feed [22,23]. Cyanobacteria are prokaryotic organisms that are sometimes included in the algae category due to their similar characteristics, behavior and habitat to microalgae, although they differ from algae in that they consist of prokaryotic cells. Also, the dietary supplementation of *Schizochytrium* spp. Goldstein & Belsky as an additive has been reported to have beneficial effects because of its high content of n-3 and n-6 polyunsaturated fatty acids [24,25].

Several products containing algae are currently available on the market for both human and animal use; however, although the beneficial effects of individual algae are known, the possible beneficial effects of using several algae in combination have not yet been fully evaluated. The combination of different species of algae can highlight complementary effects that may lead to a greater benefit than single algae for animal health when fed to animals. Although the nutritional value of algae and their positive effect on the diet has been recognized due to the large number of algal species and their different characteristics, it is necessary to evaluate their individual activities and possible synergistic and/or antagonistic effects. To investigate the possible use of these compounds as functional ingredients in pig farming, the aim of this study was to evaluate the in vitro antioxidant activity and the antibacterial effect against O138 *E. coli*, a common pathogen in pig livestock, comprised of four different algal extracts (*Ascophyllum nodosum; Chlorella vulgaris; Lithotamnium calcareum; Schizochytrium* spp.) and *Arthrospira platensis,* a cyanobacteria. Additionally, the viability and the integrity of the IPEC-J2 cell membrane was investigated in order to clarify if the addition of algae or cyanobacteria as a functional feed could lead to some changes in the intestinal epithelium.

This study will contribute to the scientific knowledge on the potential role of algal and cyanobacterial compounds in animal feed as functional additives and alternatives to antibiotics and also, a possible beneficial effect of their combination.

## 2. Materials and Methods

### 2.1. Algal and Cyanobacterial Material

*Ascophyllum nodosum* (L.) Le Jolis; *Chlorella vulgaris* var. *vulgaris* Beijerinck; *Lithothamnium calcareum* (Pallas) Areschoug; *Schizochytrium* spp. Goldstein & Belsky and the cyanobacteria *Arthrospira platensis* (Nordstedt) Gomont were commercialized and purchased by Italfeed Srl (Milan, Italy), in conformity with European safety requirements.

### 2.2. Chemical Composition

The chemical analyses were performed on the samples following the “Official Methods of Analysis” according to AOAC [26]. Analyses were conducted to determine the main nutritional components (ash, crude fiber, crude protein, ether extract). Briefly, dry matter (DM) was obtained by drying the samples in a forced-air oven at 65 °C for 24 h (AOAC method 930.15). Ash (Ash) was obtained by placing the samples in a muffle furnace at 550 °C for 3 h (AOAC method 942.05). Crude fiber (CF) was determined by the filter bag method (AOCS method Ba 6a-05) [27]. Crude protein (CP) was determined by the Kjeldahl method (AOAC method 2001.11). Ethereal extract (EE) was determined by ether extraction in the Soxtec system (DM 21/12/1998). 

### 2.3. Algae Extraction

Dried meal of *Arthrospira platensis; Ascophyllum nodosum; Chlorella vulgaris; Lithotamnium calcareum; Schizochytrium* spp. were extracted using methanol and deionized water following Gouvinhas et al. [28] with some adaptations. Briefly, 40 mg of each alga was dissolved in 1.5 mL of methanol/deionized water (50:50, *v*/*v*). The mixture was then vortexed and stirred at room temperature (RT) for 30 min. Samples were centrifuged for 15 min at 10,000 rpm at 4 °C. The supernatants were collected, filtered through a 0.45 μm syringe filter and stored at −20 °C until the analysis. Secondarily, in order to test a possible synergic or combined effect, 20 mg of dried meal from each alga were mixed with 20 mg of other algae’s dried meal. The mixtures were extracted following the same extraction protocol as previously described.

### 2.4. HPLC-Exploris-Orbitrap^®^-MS Analysis

Chromatographical separation was accomplished on the Vanquish HPLC instrument (Thermo Fisher Scientific, San Jose, CA, USA) using an Restek RP column with a programmed gradient flow of 0.1% HCOOH in water and methanol. The operative conditions were set up in order to achieve the best separation of the most important polyphenolic analytes. Exploris HRMS (Thermo Scientific, San Jose, CA, USA) was operated in both positive mode and negative mode simultaneously, and each one was performed with predetermined acquisition parameters. The full scan (FS) with resolving power 120.000 (two scan range of *m*/*z* 70–800 and 800–2500) was used for the screening and statistical evaluation of the chromatographic profiles. Full scan data-dependent acquisition (FS-dd-MS2) with resolving power 60.000 and 17.500 for FS and dd-MS2, respectively, was employed for the fragmentation of pseudo-molecular ions detected in FS mode. Fragmentation of precursors was executed with stepped, normalized collision energy (NCE) set at 20, 30 and 40 eV.

### 2.5. HPLC-Exactive-HRMS Untargeted Metabolomics Approach

The detailed untargeted metabolomic workflow that was applied here was described in our recent publication [29] with slight modifications. Briefly, the Exploris Orbitrap raw data were submitted to Compound Discoverer (CD) 3.3 software (Thermo Fisher, Waltham, MA, USA) that enabled the programmed compound identification and statistical evaluation. The procedure is based on a series of steps that are consequentially accomplished: spectra selection, alignment of retention time, the precursor ions collection consulting CD integrated databases (https://www.mzcloud.org (accessed on 10 April 2022) and https://www.chemspider.com (accessed on 10 April 2022)) and normalization of the chromatographical peak areas using quality control (QC) samples as a reference. Criteria for putative identification of metabolites identified by CD workflow were chosen as a combination of a few different assets: an mzCloud match score higher than 80% and the same identification being proposed by at least one of the following external web databases: Human Metabolome platform HMDB (https://hmdb.ca/, accessed on 10 April 2022), Kyoto Encyclopedia of Genes and Genomes (KEGG), (https://www.genome.jp/kegg, accessed on 10 April 2022), Pubchem (www.pubchem.com, accessed on 10 April 2022) or Small Molecule Pathway Database (SMPDB) (http://smpdb.ca, accessed on 10 April 2022). If the mass fragmentation pattern did not correspond to any of database’s software, manual verification of the fragmentation pattern program was achieved using ChemDraw software (https://chemdrawdirect.perkinelmer.cloud/js/sample/index.html, accessed on 10 April 2022).

### 2.6. Evaluation of Antioxidant Properties (ABTS Assay)

The antioxidant activity was tested by adopting an ABTS assay, according to Dell’Anno et al. [30]. The 2,20-azino-bis (3-ethylbenzothiazoline-6-sulfonic acid) (ABTS^•+^) radical cation was generated by the reaction of 7 mM ABTS with 2.45 mM of K-persulfate. The reaction mixture was left to stand in the dark overnight at room temperature and was used within two days. The working solution of ABTS^•+^ was diluted with deionized water to obtain an absorbance of 0.700 ± 0.02 OD at 734 nm at room temperature. First, a calibration curve was obtained using different concentrations (2000 mM, 1500 mM, 1000 mM, 500 mM, 250 mM, 0 mM) of Trolox (6-hydroxy-2,5,7,8-tetramethychroman-2-carboxylic acid) as the standard. The assay was performed using 10 µL of diluted sample added to 1 mL of working solution (ABTS^•+^). The absorbance was recorded after 6 min of incubation in the dark, and all determinations were performed in triplicate.

All the algal extracts were diluted in their solvent (distilled water plus methanol, 50:50, *v*/*v*) and tested in the following concentrations: 100 vol%, 75 vol%, 50 vol%. Algae mixtures were tested through ABTS assay without further dilution. The total antioxidant capacity after six minutes of reaction was expressed as the percentage of inhibition (PI%), according to the following equation:PI = [(AbsABTS^•+^ − Abs sample)/Abs ABTS^•+^)] × 100(1)

AbsABTS^•+^ denotes the initial absorbance of diluted ABTS^•+^ and Abs sample denotes the absorbance of the sample after 6 min of reaction. All assays were performed in technical triplicate and with three biological replicates that were meant to verify the replicability of the experiment using the same procedures, which included repeating the experiment starting from the sample extraction and repeating the test on different days.

### 2.7. Molecular Characterization of Escherichia coli

The O138 *E. coli* strain belonging to our strains collection [31] was genetically characterized for the presence of genes encoding two virulence factors: the adhesive fimbriae F18 and the verocytotoxin (VT2e). Briefly, specific oligonucleotides were designed for the detection of the B subunit of the VT2e toxin, which was responsible for binding the toxin to the intestinal cell surface before the absorption and FedF gene minor subunit of F18 fimbria essential for the binding of the enterocyte receptor (Table 1). Genomic DNA was extracted using phenol/chloroform (1:1) from an overnight culture of *E. coli* strain, and the quality of DNA was evaluated spectrophotometrically (260/280 ratio) and by agarose gel electrophoresis (1.5%) to quantify and test for the presence of RNA or degraded DNA. The presence of FedF and VT2eB genes was evaluated by polymerase chain reaction (PCR) using specific primer pairs (Table 1). PCR was performed using the following conditions: first denaturation (94 °C for 2 min); denaturation phase (94 °C for 1 min); annealing phase (55 °C for 2 min); elongation phase (72 °C for 2 min); the cycle described above was repeated 34 times. The volume of the reaction mixture was 50 µL, with 5 µL of template (bacterial DNA) added to the PCR mixture.

### 2.8. Growth Inhibition Assay

Growth inhibition assay was performed against the O138 *E. coli* strain. To perform this assay, extracts of *Arthrospira platensis; Ascophyllum nodosum; Chlorella vulgaris; Lithotamnium calcareum; Schizochytrium* spp. obtained for the antioxidant assay were filtered with a 0.22 µm syringe filter and stored at −20 °C until the analysis. A liquid culture-based growth inhibition assay with *E. coli* O138 was performed to evaluate their ability to inhibit bacterial growth. An overnight culture of *E. coli* O138 in Luria–Bertani (LB) broth was used as inoculum for the experiments. The growth inhibition assay was performed as follows: the extracts were diluted in LB liquid medium in order to obtain five different concentrations (25 vol%, 12.5 vol%, 6 vol%, 3 vol%, 1.5 vol%). 100 µL of diluted extract were added in a microtiter 96-well plate to which 30 µL *E. coli* inoculum was also added. The positive controls were prepared by adding 30 µL of *E. coli* inoculum to the solution of methanol/distilled water (50:50, *v*/*v*) in order to evaluate the bacterial growth without any external influence. To correct background color, negative controls were prepared by adding 30 µL of LB without *E. coli* inoculum. All samples were then incubated at 37 °C in a shaking incubator for six hours. The growth rate of *E. coli* was estimated every hour for six hours by measuring the absorbance with a microplate reader spectrophotometer (ScanReady P-800, Life Real, Zhejiang, China) at an optical density (OD) of 620 nm. The measured OD was converted into log_10_ of the number of cells/mL, considering 1 OD = 1 × 10^9^ cells/mL [32]. All assays were performed in technical quadruplicate and with three biological replicates that were meant to verify the replicability of the experiment using the same procedures, which included repeating the experiment starting from the sample extraction and repeating the test on different days.

### 2.9. Cell Treatment

Cells were treated with different concentrations of *Ascophyllum nodosum* and *Lithotamnium calcareum* (0.1–5 vol%). In a second set of experiments, cells were pre-treated for 3 h with or without *Ascophyllum nodosum* and *Lithotamnium calcareum* (0.1–5 vol%) in a DMEM medium. Subsequently, the cells were challenged with H_2_O_2_ (0.5 and 1 mM) for 1 h individually to induce oxidative damage.

### 2.10. Viability Assay on Intestinal IPEC-J2 Cell

IPEC-J2 cell line was used for viability assay. IPEC-J2 is a non-transformed cell line derived from intestinal porcine enterocytes isolated from the jejunum of a neonatal unsuckled piglet (ACC 701, DSMZ, Braunschweig, Germany). The IPEC-J2 cells were cultured in Dulbecco’s Modified Eagle Medium with stable L-Glutamate and Ham’s F-12 mixture (DMEM/F-12 mix) (Immunological sciences, Società Italiana Chimici, Rome, Italy), supplemented with 15 mM HEPES (Sigma-Aldrich, Milan, Italy), 5% fetal bovine serum (FBS) (Immunological sciences, Società Italiana Chimici, Rome, Italy), 1% penicillin (100 U/mL)/streptomycin (100 mg/mL) (Euroclone, Milan, Italy) and cultivated in a humid chamber at 37 °C with 5% CO_2_. Experiments were performed using IPEC-J2 at passages of 24 to 28 to ensure reproducibility.

Cell viability was determined through the quantification of mitochondrial oxidoreductase using the 3-(4,5-dimethylthiazol-2-yl)-2,5-diphenyltetrazolium bromide (MTT) assay method according to Sundaram et al. [33]. Briefly, IPEC-J2 cells at sub-confluence were pre-treated with four different concentrations (0.1 vol%; 0.5 vol%; 1.00 vol%; 5.00 vol%) of *Ascophyllium nodosum*, *Lithothamnium calcareum* and a combination of both for three hours, and then they were challenged with hydrogen peroxide (H_2_O_2_) at two different concentrations (0.5 mM and 1 mM) for 1 h incubation to induce chemical stress. The optical density (OD) was measured at 570 nm in a colorimetric plate reader (Bio-Rad, Sigma-Aldrich) and the cell viability was calculated using the formula:Cell viability (%) = [(OD_treatment_ − OD_blank_)/(OD_control_ − OD_blank_)] × 100(2)

### 2.11. Membrane Stability Assay on Intestinal IPEC-J2 Cells

Cell membrane integrity was evaluated through the cytosolic lactate dehydrogenase (LDH) released in the culture media [33] Briefly, after treating the cells with different concentrations of the combination of the extracts of *Ascophyllium nodosum* and *Lithothamnium calcareum,* 50 µL of the supernatant was taken and mixed with an equal volume of LDH buffer (CytoTox 96^®^, Promega, Madison, WI, USA) in a 96-well plate and incubated for 30 min RT in the dark. Then, the cell membrane integrity was evaluated through a colorimetric measurement in a microplate (Bio-Rad) at 490 nm.

### 2.12. Statistical Analysis

All the data were analyzed using GraphPad Prism software (Version 9.0.0). The normality distribution of data and residuals were evaluated by Shapiro−Wilk, Anderson−Darling, D’Agostino−Pearson and Kolmogorov−Smirnov tests. Homoscedasticity was assessed using Brown−Forsythe and Bartlett’s tests. For the antioxidant activity assay, the data were analyzed using one-way analysis of variance (ANOVA). Data concerning the synergistic effect on antioxidant activity were analyzed using one-way analysis of variance (ANOVA). For the growth inhibition assay, the data were analyzed using two-way analysis of variance (ANOVA), including the effect of treatment, time and their interaction. The size effect of significant differences was assessed by calculating the eta squared. The one-way analysis of variance (ANOVA) was used for the analysis of data regarding the viability and membrane integrity of IPEC-J2 cells. Post-hoc pairwise comparisons were performed through Bonferroni Sidak’s test. Data were reported as mean ± standard error and differences were considered statistically significant for *p* < 0.05.

## 3. Results

### 3.1. Chemical Analysis

In order to determine the main nutritional components (ash, crude fiber, crude protein, ether extract), the chemical composition of algae and cyanobacteria were obtained through the AOAC (2005) “Official methods of analysis”. The obtained results (Table 2) showed that, in general, all the species tested have a high content of ash and consequently, of minerals. Whereas, the other nutritional components are present in minimal quantities, with the exception of *Arthrospira platensis* and *Chlorella vulgaris*, which disclose a high percentage of proteins.

### 3.2. Evaluation of Molecules with Antioxidant Properties

Table 3 shows the metabolomic profile identified by HPLC−HRMS related to molecules with antioxidant properties. *Ascophyllum nodosum* has the highest content of polyphenols (10,598.1 ng/g) and tripeptides (32,622.4 ng/g), compared to others. In particular, among the polyphenols found in the *Ascophyllum nodosum* extract, those most present are Phloroglucinol (6554.2 ± 635.0 ng/g) and 4-Coumaric acid (2539.1 ± 181.8 ng/g).

### 3.3. Antioxidant Activity

During the evaluation of the antioxidant properties of the four algae (*Ascophyllum nodosum; Chlorella vulgaris; Lithothamnium calcareum; Schizochytrium* spp.) and *Arthrospira platensis* considered in this study, three different concentrations (100 vol%; 75 vol%; 50 vol%) of algae extract were evaluated. All the tested extracts showed a dose-dependent antioxidant effect where the percentage value of inhibition (PI%) after six minutes was significantly higher at a concentration of 100 vol% than at 75 vol% and 50 vol% concentrations for each species (*p* < 0.0001) (Appendix A).

Among all extracts, we have observed that *Ascophyllum nodosum* revealed the highest antioxidant capacity of all the concentrations tested, with a percentage of inhibition of 57.75 ± 1.44%; 49.49 ± 0.59%; 43.30 ± 1.69% (100 vol%; 75 vol%; 50 vol%, respectively) (Figure 1).

Also, the synergistic or combined effect of antioxidant activity was evaluated. The data obtained showed that above all the possible combinations listed in Appendix A, the ones that showed the highest antioxidant capacity were *Ascophyllum nodosum* and *Lithothamnium calcareum* (Figure 2). This combination displayed the highest antioxidant effect with a PI of 48.84 ± 1.37%, thus suggesting a combined effect for the combination of algal species. In fact, the sum of individual activities was lower than the activity of the combined extract (45.10 ± 1.74% and 48.84 ± 1.37%, respectively). Additionally, the combination of *Chlorella vulgaris* and *Lithothamnium calcareum* showed a combined effect, in which the activity of the combined extract was higher than the sum of the individual activities (9.59 ± 1.10% and 6.27 ± 0.24%, respectively). For the other combinations, the obtained data showed an antioxidant effect lower than that exerted by the single algae or cyanobacteria (*p* < 0.0001) (Appendix A).

### 3.4. Growth Inhibition Activity

The growth inhibition activity was evaluated for each extract considering different dilution (1:4; 1:8; 1:16; 1:32; 1:64) against O138 *E. coli*. Their antimicrobial capacity, meant as growth inhibitory activity, decreased with increasing dilution. Also, at high dilutions, especially 1:32 and 1:64 (Appendix A), the tested extract was not able to inhibit the growth of *E. coli*, showing a value of growth that is comparable to those of the positive control (*E. coli*). In particular, *Arthrospira platensis, Chlorella vulgaris;* and *Schizochytrium* spp. have been shown to have a significant antimicrobial (*p* < 0.05; eta squared = 0.0095, 0.09 and 0.14, respectively) capacity at a dilution of only 1:4 (Figure 3). *Lithothamnium calcareum* has shown an inhibition capacity against *E. coli* at a dilution of 1:4 (Figure 3), but also at a dilution of 1:8 (*p* < 0.05; eta squared = 0.02) (Appendix A). Finally, *Ascophyllum nodosum* has been observed to demonstrate inhibitory activity at three different dilutions: 1:4 (*p* < 0.05; eta squared = 0.25); 1:8 (*p* < 0.05; eta squared = 0.01); 1:16 (*p* < 0.05; eta squared = 0.006) (Figure 3, Appendix A).

Moreover, a time-related effect was observed for the inhibitory activity. In fact, the assay revealed that after the first few hours, the inhibition capacities of the tested algae were comparable to the positive control (*E. coli*) and only after at least 4 h did the difference in the inhibition capacity between the tested compounds and the positive control become significant.

### 3.5. Viability and Membrane Integrity of Intestinal IPEC-J2 Cells

*Ascophyllum nodosum* and *Lithotamnium calcareum* algal extracts showed greater growth inhibition and antioxidant properties; therefore, they were also tested in swine intestinal epithelial IPEC-J2 cells to determine whether they can also affect the viability of the cells under normal conditions and after experimentally induced oxidative stress. Dose−response curves with different concentrations of *Ascophyllum nodosum*, *Lithotamnium calcareum* and the combination of both algae were tested on IPEC-J2 cell viability. The results showed that at the highest concentration (5 vol%) of *Ascophyllum nodosum* and *Lithotamnium calcareum* tested, IPEC-J2 cell viability was significantly reduced (*p* < 0.05) when compared with the control (viability of IPEC-J2 cell without algal pre-treatment). On the other hand, the incubation with the combination of the two algal extracts disclosed that at the concentration of 0.1 vol%, the cell viability was significantly increased (*p* < 0.05). While at the other concentrations tested (0.1 vol%, 0.5 vol% and 1 vol% for the extract of *Ascophyllum nodosum*; 0.1 vol% and 0.5 vol% for the extract *Lithotamnium calcareum*; 0.5 vol%, 1 vol% and 5 vol% for the combination of the two algae extracts), the cell viability is not affected (Figure 4a).

We also tested the trophic effect of the extracts of *Ascophyllum nodosum*, *Lithotamnium calcareum* and their combination on IPEC-J2 cells experimentally stressed with H_2_O_2_. In particular, IPEC-J2 cells were pre-treated for three hours with algal extract at different concentrations and further challenged with two different concentrations of H_2_O_2_ (0.5 mM and 1 mM) for 1 h to simulate in vitro conditions of oxidative stress at the intestinal cell epithelium layer level. In the H_2_O_2_ 0.5 mM-challenged IPEC-J2 cells, the 3 h pre-treatment with the combination of both algal extracts mitigated the oxidative stress experimentally induced by increasing the cell viability. However, the pre-treatment with *Ascophyllum nodosum* and *Lithotamnium calcareum* alone does not mitigate the oxidative stress caused experimentally while leaving cell viability unaltered (Figure 4b). In the H_2_O_2_ 1 mM-challenged IPEC-J2 cells, the 3 h pre-treatment with algal extracts was ineffective in mitigating the oxidative stress caused experimentally, leaving the cell viability unaltered (Figure 4c).

Moreover, LDH assay was performed to determine the effect of algal extracts on cell membrane integrity. The results disclosed that the membrane integrity was not affected by pre-treatment with *Ascophyllum nodosum* and *Lithothamnium calcareum* placed in combination (Figure 5).

## 4. Discussion

In this study, the attention was focalized on the in vitro evaluation of antioxidant and antimicrobial activities as nutraceutical properties of four different types of algal extracts and one cyanobacterial extract suitable for animal nutrition (*Arthrospira platensis*; *Ascophyllum nodosum*; *Chlorella vulgaris*; *Lithotamnium calcareum*; *Schizochytrium* spp.) in order to establish their further use as functional additives and as a possible alternative to antibiotics.

### 4.1. Chemical Analysis

The obtained results of the chemical analysis of algae are in line with the literature [34] and the commercial feed label. *Ascophyllum nodosum* and *Lithothamnium calcareum* are characterized by a high content of minerals. This aspect should be considered in the diet‘s formulation; in fact, if algae are used as feed additives, they will be included in a percentage of less than 5% of the diet, so as not to constitute a major change in the mineral balance and respect the admitted levels of the European Union regulation (Reg 1081/2003/EC) (EC 2003). On the contrary, *Schizochytrium* spp. contains a lower amount of minerals (5.42% DM), but it represents an important source of lipids. *Arthrospira platensis* and *Chlorella vulgaris* are characterized by a high value of crude protein content with high biological value [35]. Even if the amount of protein is high, it should be considered that algae also contain non-protein nitrogen and this can slightly affect the results. In particular, the presence of non-protein nitrogen could lead to an overestimation of protein content if this is calculated using the standard conversion value for nitrogen (6.25) [36].

The analysis of the metabolomic profile of extracted samples showed that they are rich in polyphenols. Polyphenols are a large family of naturally occurring organic compounds characterized by multiple phenol units [37]. Phenolic compounds are considered one of the most important classes of natural antioxidants, so much so that the antioxidant activity is considered one of the main properties associated with them [38]. Moreover, as found in the metabolomic profile, the analyzed algae are also rich in tripeptides and, in particular, oxidized glutathione. Glutathione is a water-soluble tripeptide that plays a crucial role in the antioxidant response. In addition to its role as a cofactor in the neutralization of hydrogen peroxide, it can directly inactivate superoxide and hydroxyl radicals, as well as singlet oxygen [39]. Therefore, the presence of these compounds in algae makes them interesting functional ingredients because, in addition to providing benefits at the nutritional level, they can help counteract oxidative stress within the animals.

### 4.2. Antioxidant Activity

As the use of synthetic antioxidants has been questioned, interest in finding new antioxidant agents is growing and among these, antioxidants from natural sources appear to be promising [40,41]. Of particular interest are the natural antioxidants found in algae and their extracts. Seaweeds are known to contain reactive antioxidant molecules, such as ascorbate and glutathione (GSH) when fresh, as well as secondary metabolites, including carotenoids (α- and β-carotene, fucoxanthin, astaxanthin), mycosporine-like amino acids (mycosporine-glycine), catechins (i.e., catechin, epigallocatechin), gallate, phlorotannins (i.e., phloroglucinol), eckol and tocopherols (α-, χ-, δ-tocopherols) [42,43]. Also, microalgae have been shown to have antioxidant capacity because of their high content of fatty acids, carotenoids and phenolic compounds [44].

In our study, the antioxidant capacity was evaluated after six minutes of reaction between ABTS^•+^ and algae extracts according to a standard procedure described by Dell’Anno et al. [30], based on the timing defined for the radical cation decolorization assay. As it was shown that even if free antioxidant substances, in general, react immediately when the sample is added to an ABTS^•+^ reaction mixture (up to 3 min), other antioxidants that are not immediately available may require time to be released in order to exert their effect against the radicals (up to 6 min) [45,46].

This study demonstrated that all the considered algae showed a dose-dependent effect. In particular, at a concentration of 100 vol% the antioxidant activity of the algal extract was higher compared to that at 75 vol% and 50 vol% for each algal species. This is because by diluting the extract, the concentration of antioxidant compounds in the extracts decreases, leading to a lower antioxidant capacity of the extract.

Among all the algae tested, *Ascophyllum nodosum* showed the higher antioxidant capacity with a percentage of inhibition of 57.75 ± 1.44%. The high value antioxidant activity of *Ascophyllum nodosum* is probably due to the phlorotannin content, which possess strong antioxidant capacity [47], and the presence of laminarin and fucoidans, which are two polysaccharide groups that have been proven to have antioxidant activity [48].

*Arthrospira platensis*, commonly known as Spirulina, and *Schizochytrium* spp. in our study demonstrated a lower antioxidant activity compared to *Ascophyllum nodosum*. Our studies demonstrating the antioxidant capacity of both *Arthrospira platensis* and *Schizochytrium* spp. are in line with several in vivo studies which have largely demonstrated the capacity of both species of algae to improve the health status and production efficiency of livestock animals [49,50,51]. *Schizochytrium* spp. contains a large amount of DHA that have many beneficial effects. However, high concentrations of n-3 polyunsaturated fatty acids may increase lipid peroxidation and subsequently induce oxidative stress [52]. This may explain the lower antioxidant capacity of *Schizochytrium* spp. compared to *Ascophyllum nodosum*. The PI of 9.80 ± 1.23% and 2.97 ± 0.14% for *Chlorella vulgaris* and *Lithothamnium calcareum*, respectively, represented the lowest antioxidant capacity compared to the other algae considered in this study. Although our in vitro experiment disclosed low values of antioxidant activity, it may be due to the extraction method and the concentrations that were tested. Some in vivo studies highlighted the potential beneficial effect of adding *Chlorella vulgaris* as a nutritional additive in animal diets; in fact, this addition has been shown to positively contribute to the alleviation of oxidative stress, as well as to the strengthening of the body’s non-specific defenses and the improvement of zootechnical performance [53,54,55].

In general, the difference between the observed in vivo results and those obtained in vitro in different literature studies could be due to the disparate algae origin, cultivation conditions, extraction methods used and tested concentrations. Indeed, the extraction methods used to obtain the algae’s compounds vary, and the following can be distinguished in ethanol, methanol, enzymatic composting and supercritical CO_2_ extraction with solvents [56]. However, the effectiveness of each method is related to the species of algae, the target compounds to be extracted and environmental factors [57,58,59].

In addition to the antioxidant activity of each algal extract, a possible combined or synergistic effect was also evaluated during the study. The combined effect was evaluated on all the possible combinations compared to the algal extract obtained for each species. The data obtained disclosed that only a few algae can develop a synergistic or combined effect when extracted in combination. Only two algal combinations (*Ascophyllum nodosum* + *Lithothamnium calcareum* and *Chlorella vulgaris* + *Lithothamnium calcareum*) revealed a synergistic effect. In fact, comparing the sum of the antioxidant capacity of the individual algae extracts at a concentration of 50 vol% (43.27 ± 1.69% and 1.84 ± 0.08% of the PI of *Ascophyllum nodosum* and *Lithothamnium calcareum*, respectively) with the antioxidant capacity of their combination, it can be seen that the latter is significantly higher. Likewise, for the combination of *Chlorella vulgaris* and *Lithothamnium calcareum*, comparing the sum of the antioxidant capacity of the individual algae extracts at a concentration of 50 vol% (4.43 ± 0.18% and 1.84 ± 0.08% of the PI of *Chlorella vulgaris* and *Lithothamnium calcareum*, respectively) with the antioxidant capacity of their combination, it can be seen that the latter is significantly higher. Even considering the limited effect size observed with the obtained biological extracts, these data highlight that the sum of the antioxidant capacity of the single extracted algae was lower compared to the extracted mixture of algae, suggesting a possible combined effect. Many literature studies confirm the antioxidant capacities of considered algae and cyanobacteria species [45,60,61] and it has been demonstrated that the combination of different antioxidant sources could enhance their effect on radical scavenging activity [62,63,64]. In addition to the synergistic effect of some algae and cyanobacteria, the data we obtained showed a possible inhibitory effect of other combinations. This suggests that the co-presence in the same environment as some of the extracted bioactive compounds does not allow them to fully exhibit their functional characteristics. This may be due to the fact that the antioxidant activity should be related to the natural combination of phytochemicals and being different from each other and present in large quantities within nutrients contained in feed and foodstuffs, their bioavailability and distribution can be affected. Therefore, a combination of different substances can be inhibitory, synergistic or additive [65,66]. However, the precise mechanisms of action that allow the algal and cyanobacterial extract combinations to show an inhibitory, synergistic or additive effect still need to be clarified by further studies.

### 4.3. Growth Inhibitory Activity of O138 E. coli

Antibiotics have been widely used in animal production for the treatment of various bacterial diseases. The excessive use of antimicrobials has led to the emergence of antibiotic-resistant bacteria in animals, as well as in humans [67,68,69,70]. As a result, there has been both a reduction in the efficacy of antibiotics and an increased risk of transmission of antibiotic-resistant pathogens (AMR) to humans [69]. It is therefore necessary to reduce the use of antibiotics by finding alternatives in functional feed additives. Consequently, in our study, we decided to evaluate the antimicrobial capacity of algal extracts against *Escherichia coli*, in particular the O138 *E. coli* strain, which is one of the most frequent pathogens involved in the incidence of PWD. Coping with this pathogen is very difficult for farmers, as antibiotics are not effective against the toxins produced and for this reason, it is important to find a way to prevent the onset of diseases caused by O138 *E. coli* [71,72].

The O138 *E. coli* strain belonging to our strains collection [31] was genetically characterized in order to verify the presence of two virulence factors: the VT2e toxin and the F18 adhesive fimbriae (Appendix A). The presence of these virulence genes, as well as the presence of other virulence factors, are responsible for the pathogenicity of *E. coli* [5]. In addition, the O138 *E. coli* is one of the main enteric pathogens of weaned piglets, responsible for post-weaning enteritis and enterotoxaemia, causing significant morbidity and mortality in pigs worldwide. PWD is a major problem in pig farming. The occurrence of PWD may require treatment with antibiotics due to its negative impact on animal health, which then leads to important economic losses [31].

Although the observed growth inhibitory antimicrobial effects do not exhibit particularly high effect size values, the data obtained disclosed that *Ascophyllum nodosum* revealed the highest inhibitory effect on *E. coli* growth compared to the other algal species. *Ascophyllum nodosum* inhibitory activity was observed at three different successive dilutions (1:4; 1:8; 1:16), demonstrating the highest antibacterial activity measured among these extracts, proving effective even at a dilution of 1:16. The inhibitory effect of *Ascophyllum nodosum* against the growth of *E. coli* is probably due to some functional compounds of brown algae, such as phlorotannins, which are polyphenols known to have bacteriostatic and bactericidal activity [58,73]. Also, *Lithothamnium calcareum* showed antimicrobial activity. Compared with the growth of the positive control (*E. coli*), it showed an inhibitory capacity at both 1:4 and 1:8 dilutions. The inhibitory effect disclosed by *Lithothaminium calcareum* could be due to the potential of red algae species to produce antimicrobials metabolites, such as diterpenes [74], monoterpenes [75], phenolic compounds [76], sterols [77], polysaccharides [78] and fatty acids [79]. The other extracts tested showed a significant inhibitory capacity only at a 1:4 dilution, showing that their extract can only demonstrate antimicrobial activity at the highest concentration tested. The result obtained may also be due to the susceptibility of the O138 *E. coli* strain to the different antimicrobial substances present in the algal and cyanobacterial extracts, and the method of extraction may have also influenced the antimicrobial activity, as the presence of some compounds in the extracts may have masked the effect of the antimicrobial activity [80].

In general, our findings highlighted the need to use the highest concentrations in order to guarantee a significant antimicrobial effect against the growth of O138 *E. coli*. The observed eta squared for the growth inhibitory effect was probably limited due to the extraction method that allowed a crude extract to be obtained without particular purification or concentration steps. In fact, even if the results obtained disclosed the antimicrobial activity for all the extracts, for some of them, the concentrations tested were too low to ensure a significant growth inhibition. However, even if the inhibitory effect is able to be highlighted with only the highest inclusion of biological extracts, several literature studies confirmed the antimicrobial effect of natural extracts obtained from the considered algae and cyanobacteria species [81,82,83]. In addition, our studies revealed that the inhibitory capacity of the different extracts tested is time-dependent; in fact, after the first 3 h, the inhibitory capacities were comparable to the positive control (*E. coli*) and only after at least 4 h was an inhibitory effect starting to be observed. For *Ascophyllum nodosum* and *Chlorella vulgaris*, at the highest concentration tested (1:4), a significant inhibitory capacity could be observed after 4 h. Whereas for the other algae, a significant difference compared to the positive control (*E. coli*) could be observed only after 5 or 6 h. Moreover, the obtained results also highlighted the presence of interactions between the time and the concentration. In fact, for the algae that also displayed antimicrobial activity at concentrations lower than 1:4, the inhibitory capacity against the positive control (*E. coli*) become significant after 5 and 6 h. This may be due to the fact that as the algae extract is more diluted, it needs more time to show a significant, albeit slight, inhibitory effect.

### 4.4. Viability of Intestinal IPEC-J2 Cells

We also tested in vitro *Ascophyllum nodosum* and *Lithothamnium calcareum* on swine IPEC-J2 cell viability using a previously developed oxidative stress response model [33] to determine whether these two algal extracts, which are those that have been shown to have good antioxidant and antimicrobial properties tested alone or in combination, could also affect the viability of the intestinal cells under normal or stress conditions. The highest concentrations of algal extracts (1% and 5% for *L. calcareum* and 5% for *A. nodosum*) included in IPEC-J2 medium showed a decrease in cell viability under normal conditions.

Even using different cell line models, some studies reported a decrease in cell viability with increasing concentrations of phytochemicals [84]. It has been shown that in vitro, the increasing concentrations of polyphenols can impair the cell viability due to a possible toxic effect exerted at high concentrations [85].

In addition, only the combination of the two algal extracts was found to be effective in stimulating cell viability when administered at the lowest concentration. This can be traced to the fact that bioactive compounds with antioxidant capacity exist in combination and the combination of different antioxidants can act additively [86]. However, the polyphenolic compounds responsible for improving cell viability and their exact mechanism of action remain to be determined in further studies.

Moreover, we tested the effect of a 3 h pre-treatment of *Ascophyllum nodosum* and *Lithothamnium calcareum* algal extract alone or in combination with oxidative stress experimentally induced in IPEC-J2 cells [33]. The combination of the two algal extracts at a concentration of 0.5 vol% and 1 vol% are able to mitigate the effect of oxidative and inflammatory stress induced through the addition of H_2_O_2_ 0.5 mM (considered a mild stress condition). Thus, this highlights the potential of the combination of *Ascophyllum nodosum* and *Lithothamnium calcareum* for the prevention of oxidative and inflammatory conditions at the intestinal level. A limited number of studies have investigated the ability of algal extract to affect intestinal cell proliferation, but other studies reveal that *Ascophyllum nodosum* increases the viability of IPEC-J2 in a dose-dependent manner, proving that *Ascophyllum nodosum* can be included in the swine diet without any risk for the intestinal cells. Thus, *Ascophyllum nodosum* is a potentially functional ingredient in the swine diet [87].

## 5. Conclusions

Since algae and cyanobacteria have several agronomic and environmental advantages and are known to have several nutritional properties, they could become a valuable functional feed additive. However, due to the large number of algal species and their different characteristics, it is necessary to evaluate their individual activities and possible synergistic effects. For this purpose, we evaluated the antioxidant and antibacterial capacities of four algal and cyanobacterial extracts.

Chemical and metabolomic analyses have highlighted not only the high nutritional value of the species analyzed, but also their richness in biologically active compounds, which makes them a rich source of natural antioxidants and antimicrobial compounds. All algal extracts showed antioxidant abilities. In particular, *Ascophyllum nodosum* extract showed the highest antioxidant effect compared to other algal species. In addition, *Ascophyllum nodosum* exhibited a greater inhibition capacity on *E. coli* growth. A synergistic antioxidant effect of *Ascophyllum nodosum* and *Lithothamnium calcareum* was observed, suggesting the complementary characteristics of these algal species. Other algal extracts showed weak antioxidant and antibacterial properties. *Ascophyllum nodosum* revealed both antioxidant and antimicrobial capacity, while *Lithothamnium calcareum* was able to modulate the *E. coli* growth and the combination of these two species could enhance their antioxidant power. Moreover, *Ascophyllum nodosum* and *Lithothamnium calcareum* put in combination exhibit the ability to mitigate the effect of oxidative and inflammatory stress on pig intestinal cells. The combination of *Ascophyllum nodosum* and *Lithothamnium calcareum* also show potential in their ability to mitigate oxidative stress experimentally induced in IPEC-J2 cells.

Although the results obtained in vitro cannot be directly translated to in vivo, our study demonstrated functional characteristics that can be crucial for the improvement of animal health and the reduction of antibiotics. Therefore, these algae and cyanobacteria species can be considered interesting for swine nutrition. Therefore, further studies on this topic are needed to confirm the encouraging results observed in vitro and in breeding conditions and to evaluate further extraction methods that could improve these properties.

## Figures and Tables

**Figure 1 antioxidants-11-00992-f001:**
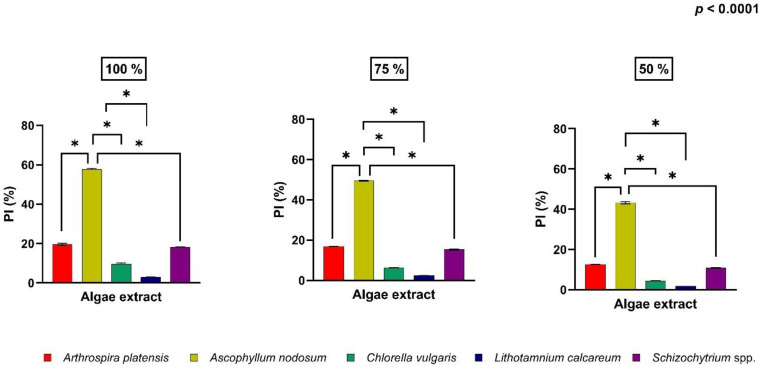
Percentage of inhibition (PI%) of radical scavenging activity at three different concentrations (100%; 75%; 50%) of four algal and cynobacterial extracts tested (*Arthrospira platensis; Ascophyllum nodosum; Chlorella vulgaris; Lithothamnium calcareum* and *Schizochytrium* spp.). Data are shown as means and standard deviations. * Asterisk indicates statistically significant differences among tested compounds (Treatment *p* < 0.0001).

**Figure 2 antioxidants-11-00992-f002:**
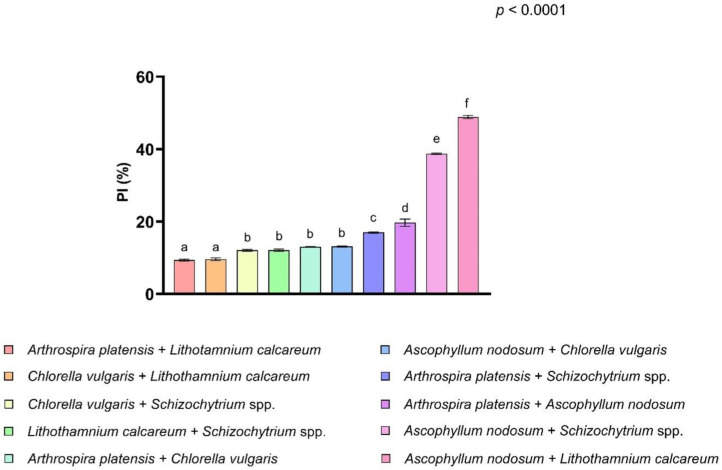
Antioxidant activity of algae and cyanobacteria combination. Figure shows the percentage of inhibition (PI%) of all the possible combinations between the species analyzed. Data are shown as means and standard deviations. All the combinations are significantly different compared to the combination of *Ascophyllum nodosum* extract and *Lithothamnium calcareum* extract. Data are shown as means and standard deviations. ^a–f^ Means (*n* = 3) with different superscripts are significantly different (Treatment *p* < 0.0001).

**Figure 3 antioxidants-11-00992-f003:**
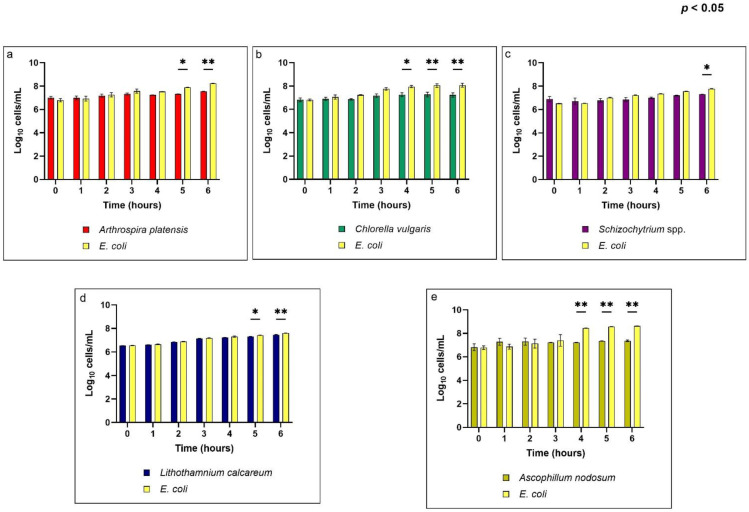
Evaluation of growth inhibition of *Arthrospira platensis*, *Chlorella vulgaris*, *Schizochytrium* spp., *Lithothamnium calcareum* and *Ascophyllum nodosum* extracts at a dilution of 1:4 against *E. coli*. (**a**) Growth inhibition of *Arthrospira platensis*. (**b**) Growth inhibition of *Chlorella vulgaris*. (**c**) Growth inhibition of *Schizochytrium* spp. (**d**) Growth inhibition of *Lithothamnium calcareum*. (**e**) Growth inhibition of *Ascophyllum nodosum*. Data are shown as means and standard deviations. Asterisk means (*n* = 3) with different superscripts are significantly different, * *p* < 0.0001, ** *p* < 0.05.

**Figure 4 antioxidants-11-00992-f004:**
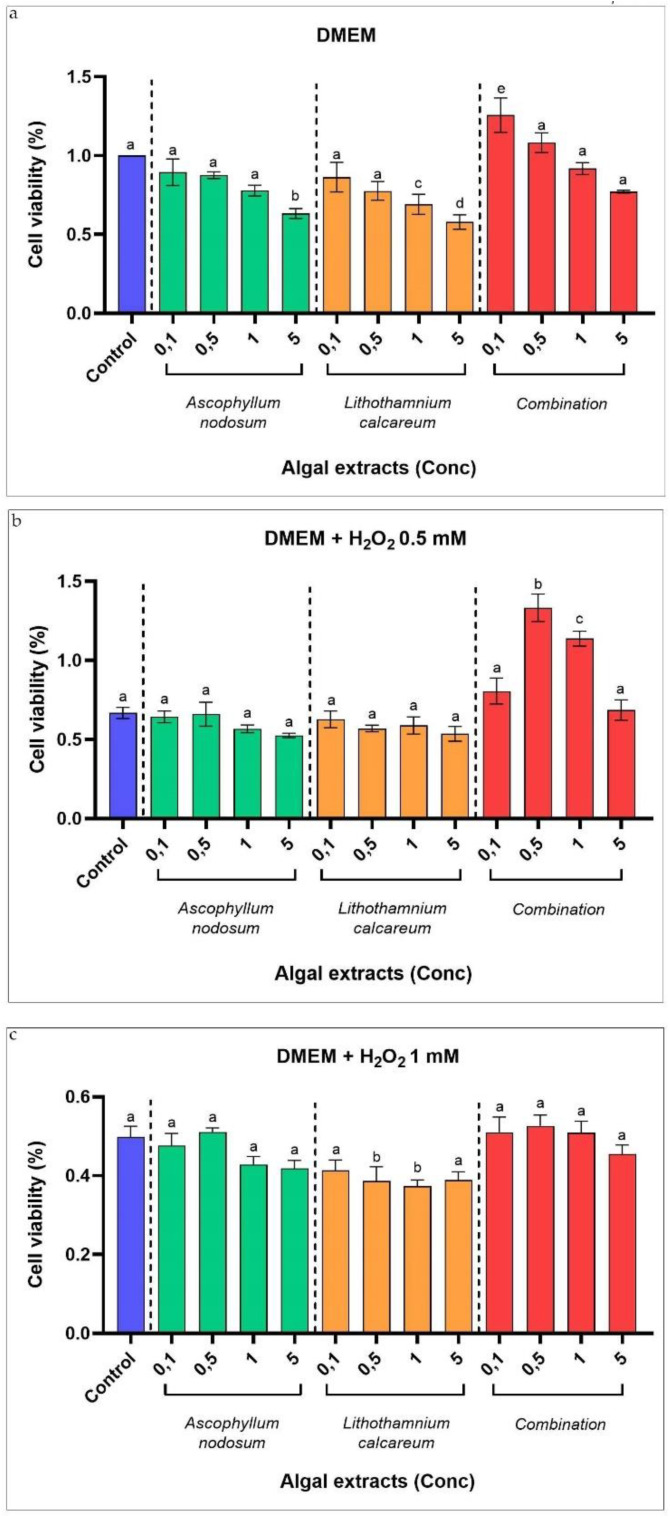
Viability of intestinal IPEC-J2 cells. (**a**) Dose−response curves with different concentrations of *Ascophyllum nodosum, Lithotamnium calcareum* and the combination of both algae; (**b**) Viability of IPEC-J2 cells challenged with H_2_O_2_ 0.5 mM and pre-treated with *Ascophyllum nodosum, Lithotamnium calcareum* and the combination of both algae; (**c**) Viability of IPEC-J2 cells challenged with H_2_O_2_ 1 mM and pre-treated with *Ascophyllum nodosum, Lithotamnium calcareum* and the combination of both algae. Data are shown as means and standard deviations. ^a–e^ Means (*n* = 3) with different superscripts are significantly different (Treatment *p* < 0.05).

**Figure 5 antioxidants-11-00992-f005:**
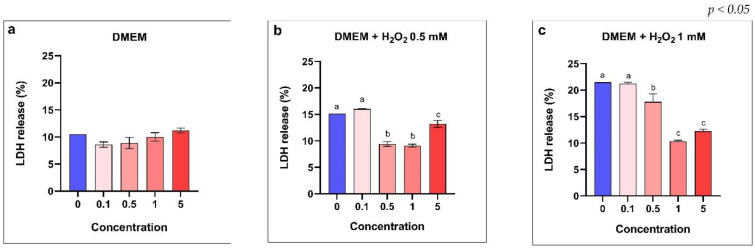
Membrane integrity of intestinal IPEC-J2 cells. (**a**) Dose−response curves with different concentrations of the combination of Ascophyllum nodosum and Lithotamnium calcareum; (**b**) Membrane integrity of IPEC-J2 cells challenged with H_2_O_2_ 0.5 mM and pre-treated with the combination of Ascophyllum nodosum and Lithotamnium calcareum; (**c**) Membrane integrity of IPEC-J2 cells challenged with H_2_O_2_ 1 mM and pre-treated with the combination of Ascophyllum nodosum and Lithotamnium calcareum. Data are shown as means and standard deviations. ^a–c^ Means (*n* = 3) with different superscripts are significantly different (Treatment *p* < 0.05).

**Table 1 antioxidants-11-00992-t001:** Primer used for polymerase chain reaction (PCR).

Primers	Nucleotide Sequences
FedF-5′(F18)	CCATGGCTACTCTACAAGTAGACAAGTCTGTTTC
FedF-3′(F18)	GAGCTCTTACTGTATCTCGAAAACAATGGGCACCG
VT2e-B subunit-5′	GGATCCATGAAGAAGATGTTTATAGCGG
VT2e-B subunit-3′	AACGGGTCCACTTCAAATGATTCTCGAG

FedF gene is a minor subunit essential for adhesion of F18 fimbriae. VT2eB is the gene codifying the B-subunit of verocytotoxin type 2 variant.

**Table 2 antioxidants-11-00992-t002:** Chemical composition of dried samples of *Arthrospira platensis*; *Ascophyllum nodosum*; *Chlorella vulgaris*; *Lithothamnium calcareum* and *Schizochytrium* spp.

	DM (%)	Ash (%)	CF (%)	CP (%)	EE (%)
*Arthrospira platensis*	95.58	7.11	0.67	62.00	0.61
*Ascophyllum nodosum*	91.44	25.33	8.92	6.93	1.79
*Chlorella vulgaris*	96.61	11.80	0.99	47.20	0.65
*Lithotamnium calcareum*	99.60	92.75	2.91	0.21	0.27
*Schizochytrium* spp.	99.20	5.42	0.18	2.62	9.06

All values are expressed as percentage of dry matter (% DM). DM: Dry matter; Ash: Ashes; CP: Crude Protein; EE: Ether Extract; CF: Crude Fibre.

**Table 3 antioxidants-11-00992-t003:** Metabolomic profile related to the content of polyphenols and tripeptides of *Arthrospira platensis*; *Ascophyllum nodosum*; *Chlorella vulgaris*; *Lithothamnium calcareum* and *Schizochytrium* spp.

Biochemical Classification	Molecules	*Arthrospira platensis*	*Ascophyllum nodosum*	*Chlorella vulgaris*	*Lithotamnium calcareum*	*Schizochytrium* spp.
**Polyphenol**	Ferulic acid	675.8 ± 68.8	520.8 ± 15.2	8282.6 ± 186.0	18.5 ± 0.0	174.1 ± 20.4
	4-Coumaric acid	1909.7 ± 73.3	2539.1 ± 181.8	7853 ± 54.7	31.6 ± 2.9	138.7 ± 3.6
	Gallic acid	13.4 ± 4.8	579.9 ± 56.8	10.5 ± 1.2	11.4 ± 3.3	19.5 ± 5.5
	4-Hydroxyphenyllactic acid	56.7 ± 30.4	127.6 ± 54.1	58.5 ± 8.5	24.9 ± 2.8	139.6 ± 135.5
	Dihydrocaffeic acid	19.4 ± 1.1	48.4 ± 7.0	13.2 ± 1.2	31.8 ± 4.6	15.0 ± 2.2
	Phloroglucionol	13.5 ± 2.1	6554.2 ± 635.0	65.2 ± 3.3	4.0 ± 1.1	2.5 ± 2.0
	Isoferulic acid	6.8 ± 0.2	23.1 ± 4.1	3.2 ± 0.3	11.0 ± 2.5	0.4 ± 0.0
	2-Hydroxy-4-(4-hydroxyphenyl)butanoic acid	17.7 ± 1.0	58.1 ± 4.1	20.9 ± 0.2	39.6 ± 7.4	42.9 ± 12.2
	Sorbicillin	36.6 ± 16.5	146.9 ± 96.2	129.4 ± 15.9	26.3 ± 3.6	139.7 ± 4.6
	2,6-Diphenylphenol	ND	ND	37.6 ± 1.3	ND	ND
**Tripeptide**	Oxidized Glutathione	40.5 ± 0.4	32,622.4 ± 2004.0	1168.1 ± 213.2	158.7 ± 39.5	64.9 ± 61.1

All values are expressed as ng/g. Data are shown as means and standard deviations.

## Data Availability

The data presented in this study are available in article and Appendix A.

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
