# Peer review of "Antioxidant and Antimicrobial Activity of Algal and Cyanobacterial Extracts: An In Vitro Study"

_antioxidants, 2022, doi:10.3390/antiox11050992_

Round 1

Reviewer 1 Report

The manuscript "Antioxidant and antimicrobial activity of algal extracts: an in vitro study" is devoted to study the various properties of algae, such as synergistic antioxidant effect, antimicrobial activity, antiradical activity, the ability to mitigate the oxidative stress. Chemical composition of algae under consideration was studied with using various methods (HPLC-Exploris-Orbitrap®-MS analysis, HPLC-Exactive-HRMS, ash, crude fiber, crude protein, ether extract determinations). It was shown that Ascophyllum nodosum revealed the highest inhibitory effect in ABTS-test and in the experiments with E.coli.  A synergistic antioxidant effect of Ascophyllum nodosum and Lithothamnium calcareum was observed. This work is interesting for biochemists and food technologists.

A lot of data are presented in the manuscript. It is written clearly, well-structured and has a good scientific soundness. I think the manuscript may be published in the Antioxidants journal after minor revision after taking into account some of the remarks described below 

  1. Lines 172, 212, 228, 229, 246 etc.: Which percentage did you use – mass or volume %? It should be mentioned.

Author Response

The manuscript "Antioxidant and antimicrobial activity of algal extracts: an in vitro study" is devoted to study the various properties of algae, such as synergistic antioxidant effect, antimicrobial activity, antiradical activity, the ability to mitigate the oxidative stress. Chemical composition of algae under consideration was studied with using various methods (HPLC-Exploris-Orbitrap®-MS analysis, HPLC-Exactive-HRMS, ash, crude fiber, crude protein, ether extract determinations). It was shown that Ascophyllum nodosum revealed the highest inhibitory effect in ABTS-test and in the experiments with E.coli.  A synergistic antioxidant effect of Ascophyllum nodosum and Lithothamnium calcareum was observed. This work is interesting for biochemists and food technologists.

A lot of data are presented in the manuscript. It is written clearly, well-structured and has a good scientific soundness. I think the manuscript may be published in the Antioxidants journal after minor revision after taking into account some of the remarks described below 

  1. Lines 172, 212, 228, 229, 246 etc.: Which percentage did you use – mass or volume %? It should be mentioned.

Thank you for the time that you dedicated to our manuscript. The paper was modified following your suggestion. In particular, the percentages used are calculated on volume, this has been specified within the manuscript.

Reviewer 2 Report

The species names should be complete, e.g. Ascophyllum nodosum (L.) Le Jolis, Lithothamnion calcareum (Pallas) Areschoug, etc. “Ascophillium Nodosum” should have the second term in lower case letters (not capitalized, as is often throughout the paper). The authors should also clarify how the identification of the five species was carried out.

Because among the species investigated is also Arthrospira platensis, which is a cyanobacteria, the title should be modified to reflect the fact that not only algae are the object of the paper.

Line 158: “ChemDrow” (probably ChemDraw) should be appropriately cited.

Lines 261-262: it is not normality of the data that should be assessed for ANOVA, but rather the normality distribution of the residuals. For small number of points (as it is likely in this study), an inferential test like Shapiro-Wilk will probably be underpowered.

Line 291: “Phloroglucionol” should probably be corrected to “phloroglucinol”.

Lines 319-330: the authors discuss a number of apparently synergistic effects, but they do not discuss the antagonistic effects involving A. platensis: all combined algal extracts apparently had lower inhibition effects than the sum of their single effects. The same seems to be true for Schizochytrium spp., but with smaller antagonistic effect sizes.

Lines 358-373: as shown by Figure 4, the antimicrobial effect, if any, is very small. The authors should have used effect size metrics (such as eta squared, Cohen’s f etc), but it seems very likely that the effect size is negligible, even for the measurements performed at 6 hours.

Lines 459-461: That sentence just enunciate the myth of the natural: in reality, the fact that a substance is natural is no guarantee that is non-toxic, as the reverse is also true: the fact that a substance is man-made is not necessarily toxic. Therefore, a more sober and scientific way of formulating that sentence should be along the lines of: “The safety of the currently used synthetic antioxidants have been questioned and therefore there is an interest for finding new, safer antioxidant agents; natural sources seem promising in this respect.”

Lines 548-565: this paragraphs is treated as if the algae tested had a strong antimicrobial effect when, in fact,  the effect size observed is very minor. The small effect size should be acknowledged, and this raises high doubts about the real usefulness of these algae as antimicrobial agents. If one adds to the small effect size the negative impact on cell viability, little utility could be claimed.

Author Response

Thank you for your comments that helped us to improve the quality of our manuscript, we have revised the manuscript according to your comments. You will find the complete point-by-point response to your comments in the attached document.

Reviewer 3 Report

The article by Frazzini et al, describes the antioxidant and antimicrobial activity of 5 different algal extracts. The article is quite well organized, but it needs to be revised before being accepted for publication.

I have some suggestions for the authors:

  • it is nowhere described how the 5 algae were grown (medium, light condition, temperature, for how long...), this should be added in the materials and methods section
  • the data presented in figure 4 and 5 is this really an antimicrobial acyivity? Are the cells dying? This for me is only growth inhibition, antimicrobial activity should be tested differently.
  • table S4 and Figure 1 report the same information. I suggest to keep Figure 1 and to place table 4 in the supplementary section
  • table 5 should become S5 and I suggest keeping only Figure 2, as also in this case the info reported are reduntant. Moreover, the order of the samples should be the same between table and figure. Is the combination of A. nodosum and L. calcareum really synergistic? I can see that is slightly higher then the the two singles, but is this difference really significant? What should be the advantage of going from 43% (50%) A. nodosum and 48.8 % of the combination?
  • please combine figure 4 and 5, as they report the exact type of experiment with also the same dilution
  • the characterization of E.coli, section 3.4, breaks the overll flow of the paper. I suggest to move this part in the supplementary material, and to integrate the paragraph in the discussion on E.coli in paragraph 4.4
  • Figure S1, S2, S3 and S4 do not have any legends. I can imagine what they represent, but I might be worng. I suggest to build a unique supplementary file that contains all the images and text
  • line 409, how can the authors discuss about oxidative stress when the parameter measured os cell viability? To talk about oxidativre stress, ROS concentration should, for example, be measured.
  • check the presentation of figure 6, some paneles have borders, some not, the letters on top are not readable. I suggest to line the three figures vertically one after the other and to enlarge them
  • figure S5 should be in main text, the 3 panels should be horizontally presented to maximize space, as figure 1
  • in the discussion it is underlined that the ABTS is done for 6 minutes....but if some reactions do require time, how can you be sure that 6 minutes are enough?
  • Figure 6, how can the combination increase cell viability , while the effect is inhibiting when the extracts are given alone? the effect might not be significant for all the conditions, but it still strongly visible
  • considering the overall results is the use of A. nodosum or L. calcareum feasible? How should you further proceed to decide this?

Author Response

(The authors gave the same response as above.)

Round 2

Reviewer 2 Report

The manuscript is now definitely improved and can be published. I would still have a suggestion for the authors with respect to the statistical analysis: they report assessing normality of the residuals and have provided in their response a QQ-plot and a "residual plot". However, the latter indicates some heteroscedasticity and the authors provide no information on the evaluation of homoscedasticity.

Author Response

We thank you in advance for your valuable suggestion that allowed a significant improvement of our manuscript. You will find our response to your comment in the attached file
